# Species-Wide Phylogenomics of the *Staphylococcus aureus Agr* Operon Revealed Convergent Evolution of Frameshift Mutations

Vishnu Raghuram,[a] Ashley M. Alexander,[b] Hui Qi Loo,[c] Robert A. Petit III,[d] Joanna B. Goldberg,[e] Timothy D. Read[d]

[a]Microbiology and Molecular Genetics Program, Graduate Division of Biological and Biomedical Sciences, Laney Graduate School, Emory University, Atlanta, Georgia, USA

[b]Population Biology, Ecology, and Evolution Program, Graduate Division of Biological and Biomedical Sciences, Laney Graduate School, Emory University, Atlanta, Georgia, USA

[c]Department of Biology, Emory University, Atlanta, Georgia, USA

[d]Division of Infectious Diseases, Department of Medicine, Emory University, Atlanta, Georgia, USA

[e]Division of Pulmonary, Allergy and Immunology, Cystic Fibrosis, and Sleep, Department of Pediatrics, Emory University School of Medicine, Atlanta, Georgia, USA

**ABSTRACT** *Staphylococcus aureus* is a prominent nosocomial pathogen that causes several life-threatening diseases, such as pneumonia and bacteremia. *S. aureus* modulates the expression of its arsenal of virulence factors through sensing and integrating responses to environmental signals. The *agr* (accessory gene regulator) quorum sensing (QS) system is a major regulator of virulence phenotypes in *S. aureus*. There are four *agr* specificity groups each with a different autoinducer peptide sequence encoded by the *agrD* gene. Although *agr* is critical for the expression of many toxins, paradoxically, *S. aureus* strains often have nonfunctional *agr* activity due to loss-of-function mutations in the four-gene *agr* operon. To understand patterns in *agr* variability across *S. aureus*, we undertook a species-wide genomic investigation. We developed a software tool (AgrVATE; https://github.com/VishnuRaghuram94/AgrVATE) for typing and detecting frameshift mutations in the *agr* operon. In an analysis of over 40,000 *S. aureus* genomes, we showed a close association between *agr* type and *S. aureus* clonal complex. We also found a strong linkage between *agrBDC* alleles (encoding the peptidase, autoinducing peptide itself, and peptide sensor, respectively) but not *agrA* (encoding the response regulator). More than 5% of the genomes were found to have frameshift mutations in the *agr* operon. While 52% of these frameshifts occurred only once in the entire species, we observed cases where the recurring mutations evolved convergently across different clonal lineages with no evidence of long-term phylogenetic transmission, suggesting that strains with *agr* frameshifts were evolutionarily short-lived. Overall, genomic analysis of *agr* operon suggests evolution through multiple processes with functional consequences that are not fully understood.

**IMPORTANCE** *Staphylococcus aureus* is a globally pervasive pathogen that produces a plethora of toxic molecules that can harm host immune cells. Production of these toxins is mainly controlled by an active *agr* quorum-sensing system, which senses and responds to bacterial cell density. However, there are many reports of *S. aureus* strains with genetic changes leading to impaired *agr* activity that are often found during chronic bloodstream infections and may be associated with increased disease severity. We developed an open-source software called AgrVATE to type *agr* systems and identify mutations. We used AgrVATE for a species-wide genomic survey of *S. aureus*, finding that more than 5% of strains in the public database had nonfunctional *agr* systems. We also provided new insights into the evolution of these genetic mutations in the *agr* system. Overall, this study contributes to our understanding of a common but relatively understudied means of virulence regulation in *S. aureus*.

**KEYWORDS** *Staphylococcus aureus*, *agr*, bioinformatics, convergent evolution, genomics, hemolysis, mutation rate, phylogenetics, quorum sensing

Address correspondence to Timothy D. Read, tread@emory.edu.

The authors declare no conflict of interest.

$S$ taphylococcus aureus is a ubiquitous nosocomial pathogen that continues to plague health care settings and threaten public health. The CDC reported that 119,247 *S. aureus* bloodstream infections and 19,832 deaths occurred in the US in 2017 (1). *S. aureus* causes a wide range of diseases, such as pneumonia, osteomyelitis, endocarditis, and skin infections (2). To elicit such diverse types of infections, *S. aureus* must be able to recognize environmental cues and adapt to its microenvironment (3, 4). The *agr* (accessory gene regulator) quorum-sensing (QS) system is a key switch that links environmental sensing and virulence in *S. aureus* (5). The *agr* operon comprises two divergent promoters, P2 and P3, each driving the four genes essential for QS (*agrBDCA*) and a small RNA (RNAiii), respectively (Fig. 1A) (6, 7). AgrD is a precursor protein that is processed by the membrane-bound peptidase AgrB, into the autoinducer peptide (AIP). The secreted AIP is then recognized by a classical two-component regulatory system: AgrC, a histidine kinase, and AgrA, a response regulator that transcriptionally activates P2 and P3, thereby continuing the autoinduction. *S. aureus* has been found to have four *agr* specificity groups, each with a distinct autoinducer peptide sequence. Other *Staphylococcus* species have their specific *agr* autoinducer peptides (5, 8–10). Autoinduction and activation of P2 and P3 leads to *S. aureus* upregulating a large arsenal of extracellular toxins, such as phenol soluble modulins, hemolysins, and leukotoxins. The *agr* system also downregulates factors that facilitate cell-cell attachment, biofilm production, and immune evasion (11–15). Collectively, these sum to a cell-density-dependent switch between adherent and virulent modes.

Although the *agr* system regulates many important pathogenesis-related functions, many clinical isolates with impaired *agr* activity have been reported, which, in some conditions, may lead to worse patient outcomes (16–21). Typically, *agr*$^+$ strains produce factors that are associated with increased virulence and attenuated expression of these *agr*-mediated virulence factors seemingly lead to reduced disease severity and decreased host cell damage (22–24). While this appears paradoxical, there are several, non-exclusive, speculations as to why strains may have evolved nonfunctional *agr* systems (25). QS systems are 'public goods' that promote evolutionary cheating strategies (26–29). Impaired *agr* activity may be related to intraspecies-interspecies competition between strains with different *agr* groups, which tend to suppress each other with no obvious effect on colonization ability (30–34). Defective *agr* function may provide pleiotropically selected phenotypes, such as decreased susceptibility to vancomycin (35, 36). *agr*$^-$ strains may have traded their ability to produce energetically expensive virulence factors but may be limited in their ability to compete with the host immune system (37–39). This attenuated toxicity appears to be inconsequential or sometimes even beneficial in chronic diseases, such as cystic fibrosis, bacteremia, and osteomyelitis, where *agr*$^-$ *S. aureus* may show increased persistence and higher mortality rates compared to *agr*$^+$ strains (20, 21, 24, 31, 40). Phase variable *agr*$^-$ mutants may exist in environments that fluctuate between selection for toxicity and persistence (41). However, reduced/lack of expression of *agr* mediated virulence may be detrimental to colonization in some other circumstances (42).

It has been proposed that *agr*$^-$ strains have sacrificed long-term viability through successful between-host transmissions for increased adaptation to specific environmental niches within the host (40, 43–45). However, studies to date have typically focused on small numbers of strains in limited clinical settings. It is important to assess whether specific patterns of variation in *agr* can be observed from a genome-wide scale spanning multiple clonal lineages of *S. aureus* because it is crucial for understanding the mechanisms driving virulence regulation. The amount of publicly available genome sequences of *S. aureus* has grown rapidly over the past decade (46), offering the opportunity to examine the evolution and diversity of critical virulence determinants in *S. aureus* from a species-wide standpoint.

In this study, we developed a bioinformatics pipeline (https://github.com/Vishnu Raghuram94/AgrVATE) for rapid identification of the *agr* group from a given *S. aureus* genome as well as putative null mutations in the *agr* operon. We then used it to analyze the 42,999 *S. aureus* genomes from the Staphopia database of consistently assembled and annotated public genome sequences compiled in 2017 (46). We found that there was a high degree of purifying selection for specific alleles of *agr* genes based on *agr* group and

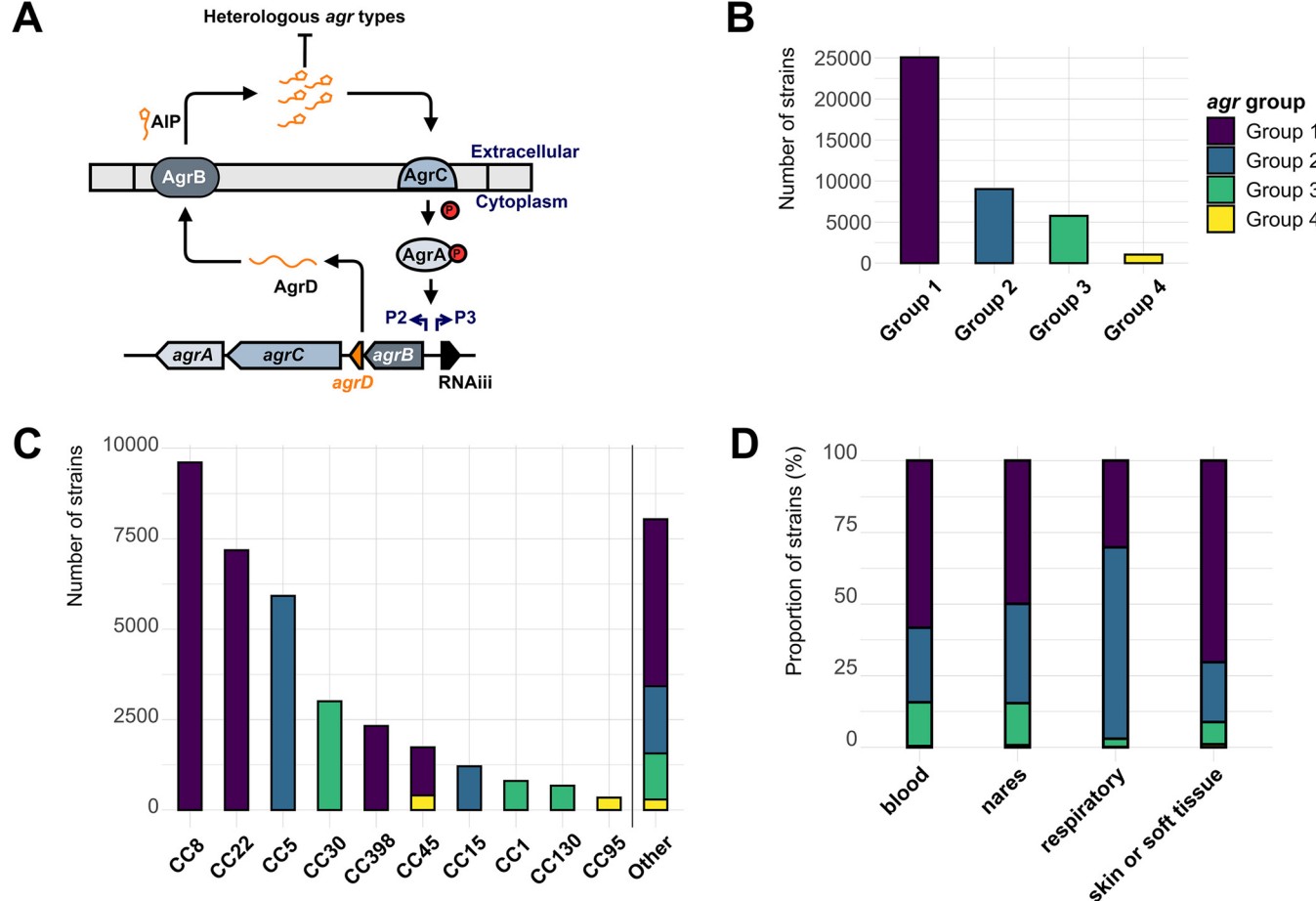

**FIG 1** Distribution of *agr* groups across 40,890 *S. aureus* genomes from the Staphopia database. AgrVATE was used to assign the *agr* groups and genomes with unknown *agr* groups were filtered out. (A) A schematic depiction of the *agr* operon showing two divergent promoters (P2 and P3) driving *agrBDCA* and small RNA RNAiii. (B) Frequency of each *agr* group in the Staphopia database. (C) Frequency of *agr* groups across the major clonal complexes (CC) of *S. aureus*. (D) Relative proportions of *agr* groups from *S. aureus* isolated from different body sites in percentage.

clonal complex. We detected frameshift mutations in 5.7% of the analyzed *agr* operons, most of which were singular events. We also detected instances of identical frameshift mutations in *agr* genes of unrelated strains across different clonal complexes, suggesting that there was a mechanism promoting mutations at these specific sites. Overall, these results highlighted the highly variable nature of the *agr* operon and suggested conserved mechanisms for acquiring genetic changes that may affect *agr* mediated virulence regulation.

The following were definitions used in this study. Group-1, group-2, group-3, and group-4 refer to *S. aureus* strains that belong to one of the four *agr* specificity groups. A cluster refers to a collection of an *agr* gene where either the nucleotide or amino-acid sequence of the gene is 100% identical among all sequences within that collection. A cluster representative was a random sequence chosen to represent each cluster whose sequence was identical to all other sequences within that cluster. A nucleotide sequence cluster representative was referred to as an allele. An amino acid sequence cluster representative was abbreviated to AACR. Frameshift mutations in the coding regions of the *agr* operon were referred to as "putative *agr* null" mutations because the true phenotype was unknown. The strains that had impaired *agr* activity were referred to as *agr*− while strains with canonical *agr* activity were referred to as *agr*+.

## RESULTS

**AgrVATE is a tool for the kmer-based assignment of *agr* groups and *agr* operon frameshift detection.** We designed the AgrVATE (*agr* variant assessment and typing engine) bioinformatic workflow to process *S. aureus* genome sequences to assign *agr* groups

and detect frameshift mutations in the *agr* operon. Current methods for *agr* group assignment involve traditional PCRs or alignment searches against *ad hoc* databases (42, 47–49). AgrVATE was designed to be a fast, standardized workflow for assigning *agr* groups that is conveniently installable through the Conda package manager (50). AgrVATE contains a database of 4 distinct collections of kmers where each collection corresponds exclusively to a single *agr* group. This kmer database was used to perform a BLASTn search against a given input genome to assign the *agr* group. The process of building and verifying this kmer database is outlined in Material and Methods. AgrVATE then extracts the *agr* operon by *in silico* PCR using usearch (51) and performs variant calling using Snippy v4.6 (52) to detect putative *agr* null mutations, such as frameshifts and early stops. As a reference for the variant calling, the cluster representative from the largest cluster for each *agr* group was used. The advantage of *in silico* PCR over global alignment methods for extracting the *agr* operon was that if the *agr* operon contains large indels/possible novel sequences which would normally break alignments, the operon was still extracted because we rely on the primers "binding" to the up and downstream regions of the operon. AgrVATE analysis took <4 s per whole-genome assembly on a Linux server with 12 core CPUs and 96 GB RAM. The AgrVATE workflow is outlined in Fig. S1.

We ran AgrVATE on 91 *S. aureus* genomes that had been typed for hemolysis activity, a phenotype that is generally associated with functional *agr* systems (16). These 91 genomes included clinical samples taken from cystic fibrosis patients from the Emory Cystic Fibrosis (CF) Center (53). We found 15 genomes had putative *agr* null mutations, 14 of which tested negative for sheep blood hemolysis (Table S1). The one putative null that displayed hemolysis on sheep blood agar had a frameshift mutation in the C-terminal end of *agrC*. The hemolysis phenotype of this strain was relatively weak (CFBR_17 in Fig. S2). We also observed 10 samples that were negative for hemolysis despite having no frameshift mutations in the *agr* operon (Table S1), suggesting that other genetic factors reduced hemolysis activity and *agr* frameshifts were not the sole indicator. In patient CFBR311, AgrVATE found two different *agr* groups (group-1 and group-2) from the genome sequence of the *S. aureus* population isolated from CF sputum (CFBR_EB_Sa110 in Table S1), showing that this patient was colonized by *S. aureus* of heterologous *agr* groups. Genome sequences of 8 individual colonies (CFBR_EB_Sa111 to CFBR_EB_Sa118 in Table S1) from this patient showed an *agr* group-2 majority (6 out of 8) and an *agr* group-1 minority (2 out of 8), thereby validating the AgrVATE prediction. This illustrated one advantage of using a kmer-based approach because AgrVATE could identify the presence of multiple *agr* groups. Each *agr* group assignment was also scored, thereby indicating the proportions of each *agr* group if more than one was present. Collectively, these findings demonstrated a potential use-case for AgrVATE in clinical settings where we identified CF patient isolates having *agr* mutations and showed one instance where AgrVATE identified a patient colonized by *S. aureus* of heterologous *agr* groups.

**Agr type distribution in the Staphopia database.** AgrVATE identified the *agr* groups for 42,491 genomes out of 42,999 in the Staphopia database with 25,539 group-1 (60.10%), 9639 group-2 (22.68%), 6224 group-3 (14.65%), and 1,089 (2.56%) group-4 genomes (Fig. 1B). Each clonal complex (CC) contained only one *agr* group, except CC45 which had both group-1 and group-4 *agr*, as had been reported previously (54, 55) (Fig. 1C). 1,601 out of the 42,491 genomes showed the presence of more than one *agr* group (Fig. S3). However, all these genomes had the secondary *agr* group call on short, low coverage contigs and there were no instances of multiple *agr* group calls on the same contig. This also showed that AgrVATE can identify *agr* groups in fragmented genome assemblies where the *agr* genes were broken across multiple contigs. These genomes were considered to be contaminated by sequences from *S. aureus* of other *agr* groups and were not included for further analysis (Fig. S3), leaving 40,890 genomes with high confidence *agr* group assignments. From the limited number of strains with the associated metadata, we found the distribution of *agr* groups across blood (3,755 genomes), skin (2,602 genomes), and nasal (4,257 genomes) isolates were similar to that of the overall distribution of *agr* groups. Group-2 genomes (CC5) were enriched in respiratory tract isolates (1,107 genomes) (Fig. 1D, Chi-squared $P < 0.01$).

The remaining 508 genomes were reported to be low confidence/unknown *agr* group calls by AgrVATE. We used a mash sketch (56) built from all publicly available complete genomes of *Staphylococcus* species to determine if these 508 genomes were indeed *S. aureus*. We found that 312 genomes did not belong to the *S. aureus* species and were likely misanno-tated submissions in NCBI and, therefore, discarded from this analysis. From the remaining 196 samples, we found samples with complete *agr* deletions (63 genomes), samples with their *agr* operons fragmented across low-quality contigs, which led to unreliable base calls (82 genomes), and samples with *agr* group-1 operons that had relatively low sequence identity (<96%) to a canonical *S. aureus agr* group-1 (51 genomes). Upon further investigation by BLASTn, we found 35 of these 51 operons belonged to *S. argenteus* species while the remaining 16 were *S. aureus* operons. The fate of all 42,999 genomes in the Staphopia database after processing them through AgrVATE is outlined in Fig. S3.

**Agr cluster recombination within clonal complexes was rare.** A total of 39,174 complete *agr* operons were extracted by AgrVATE from 40,890 genomes in the Staphopia data-base and clustered with 100% nucleotide identity, resulting in a total of 5,143 unique *agr* op-eron sequences. The remaining 1,716 genomes had their *agr* operon sequences fragmented across multiple contigs and therefore were not included for further analyses. A total of 97% of all extracted operons were of length 3481 to 3484 bp. While a single CC could harbor multiple *agr* operon clusters, no agr operon cluster was shared between genomes of differ-ent CCs, which would be expected to be produced by recombination events that intro-duced entire *agr* operons from one CC to another. When individual genes were clustered at the 100% identity threshold, we found 1,086 unique *agrA*, 440 *agrB*, 2,544 *agrC*, and 51 *agrD* alleles. As expected, there was remarkably low sequence variability in *agrD* and each allele represented a single *agr* group. All alleles of *agrB* and *agrC* were exclusive to a particular *agr* group (Fig. 2A). Across the 5143 unique *agr* operon sequences, we observed an average within-*agr* group SNP distance of 15 and between-*agr* group SNP distance of 167.

With two exceptions, there was little evidence of *agr* recombination between CCs. In the first exception, an *agrB* allele found in 1027 CC15 genomes was also found in 244 CC5 and 150 CC12 genomes. A different *agrB* allele was also found in 20 CC15 and 17 CC5 genomes. This suggested that *agr* gene alleles can be shared between CCs of the same *agr* group, although it was relatively rare. Second, when comparing the *agr* alleles across group-1 and group-4 CC45 genomes, we found that both *agr* groups had the same *agrA* allele but different group-specific *agrBDC* alleles. Specifically, out of 1,686 CC45 genomes in the Staphopia database, 1,294 were group-1 and 392 were group-4. 94% of all CC45 genomes, which included 1,217 group-1 and 384 group-4 genomes have identical *agrA* alleles. However, each *agr* group had distinct *agrBDC* al-leles, differing by an average of 179 SNPs. This suggested that a recombination event led to the stable introduction of group-4 specific *agrBDC* alleles in CC45.

Although most *agrA* alleles were CC-specific, there were multiple instances of the same *agrA* allele being found in different CCs and different *agr* groups. This lack of *agr* group specificity in *agrA* became more apparent while analyzing the amino acid sequences. A total of 83% of the AgrA amino acid sequences were identical and, therefore, had the same amino-acid sequence cluster representative (AACR). CC5, CC8, CC30, and CC45 exclu-sively had this major AACR of AgrA, encompassing all four *agr* groups (Fig. 2A, red arrow; Fig. 2B). A total of 9% had an alternate amino acid sequence of AgrA, which differed from the major AACR by a single amino acid (K136R) (Fig. 2A, blue arrow; Fig. 2B). This included mainly CC15 and other rare CCs. We designated the major AgrA AACR AgrA$^{K136}$ and the minor AgrA AACR AgrA$^{R136}$. Upon constructing a maximum likelihood phylogeny using IQ-TREE (57) from a curated set of 334 genomes from the Staphopia database each represent-ing a unique ST, called non-redundant diversity (NRD) set (46), we found that *S. aureus* could be broadly divided into two clades or subspecies. All genomes harboring AgrA$^{R136}$ were lim-ited to one clade of *S. aureus* (Fig. 2C, blue tips, blue highlighted clade). The remaining rare AgrA amino acid sequences (6%) were variants of one of the two major AACRs. To observe the linkage between individual *agr* genes, we identified SNPs in the region comprising the *agr* operon and the 1000 bp flanking regions on our filtered NRD set of 334 genomes using

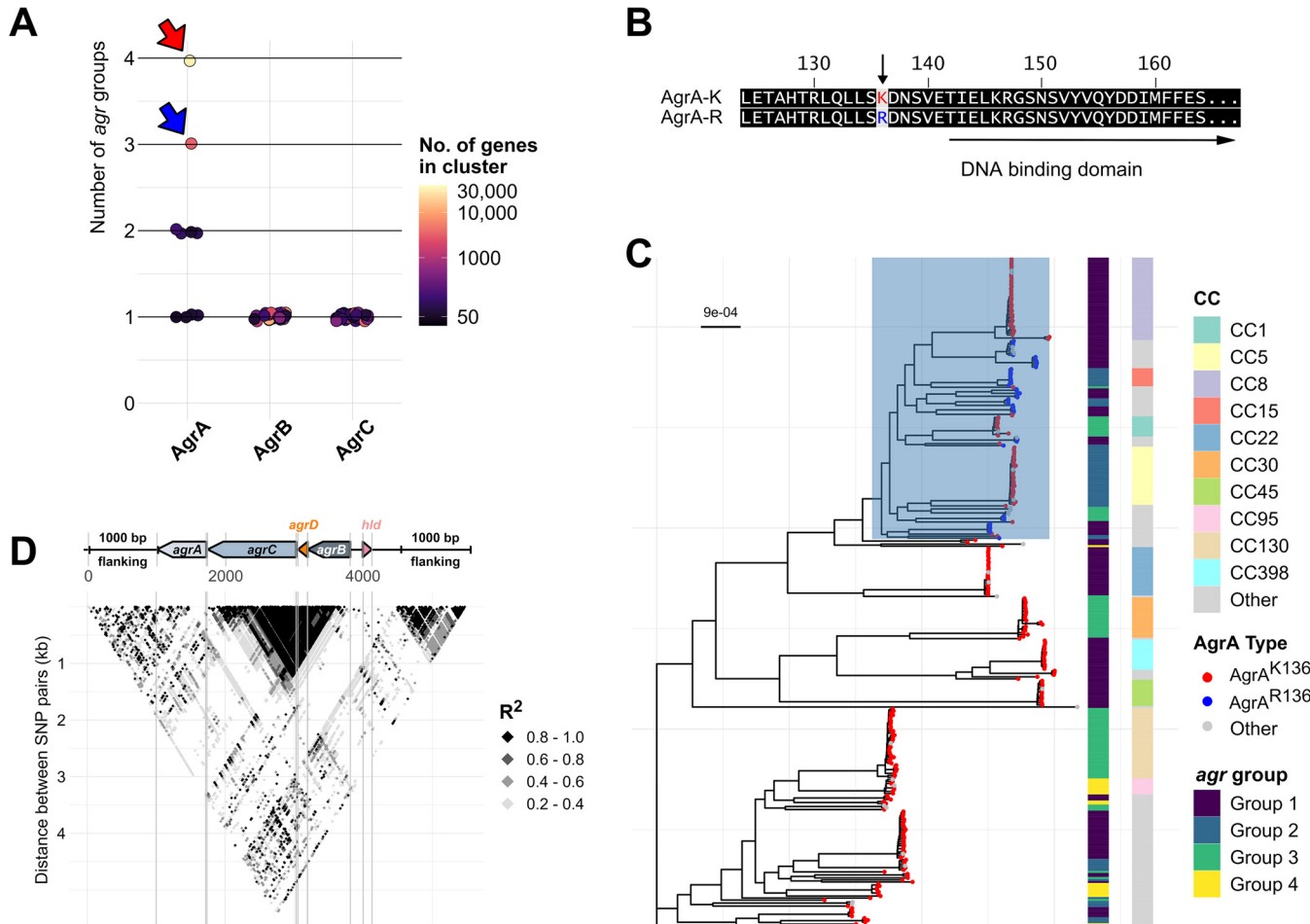

**FIG 2** AgrA evolves independently of *agr* group with only two major amino acid sequence configurations across *S. aureus*. (A) Scoring *agr* group exclusivity in clusters of unique AgrABC amino acid sequences. Extracted amino acid sequences of each *agr* gene were clustered with 100% identity to obtain all possible AA sequence configurations. Each cluster was then scored based on the number of *agr* groups the cluster sequence was found in (1 = one *agr* group; 4 = four *agr* groups) represented by a circle. Only clusters with more than 50 sequences are shown. The color of each circle represents the number of sequences within the cluster. The red and blue arrows indicate the major (AgrA$^{K136}$) and minor clusters (AgrA$^{R136}$) of AgrA AA sequences, respectively. (B) Amino acid sequence alignment of the two major alleles of AgrA. (C) Maximum likelihood phylogeny (GTR+FO model, 1000 ultrafast bootstrap replicates with average bootstrap support of 97.8%) of 334 *S. aureus* strains with each tip representing a unique ST. Tip colors represent the AgrA alleles and the corresponding heatmaps show the *agr* group and clonal complex of each tip. Scale bar indicates the number of substitutions per site. All tips representing the AgrA$^{R136}$ allele are confined to the clade highlighted in blue. (D) Linkage disequilibrium (LD) block plot of the *agr* operon and 1000 bp flanking regions. Each point on the block indicates R$^2$ values of LD calculated by plink for a given pair of SNPs. The *y*-axis indicates the distance between SNP pairs.

Snippy (52). We then measured linkage disequilibrium (LD) between these SNPs by calculating the Pearson coefficient (R$^2$) using plink (58). SNP pairs with R$^2$ >0.8 were in LD. The resulting LD plot (Fig. 2D) showed that SNPs in the variable region of *agrBDC* are in LD with each other but not with *agrA*, and *agrA* was in LD with the flanking regions of the operon. Overall, this suggested that *agrBCD* coevolve and were unlinked to *agrA* or the rest of the genome while *agrA* evolution was linked to the *S. aureus* genome.

**Nonfunctional *agr* operons are common across diverse *S. aureus* genomes.** In 39,174 *agr* operons, 405 sites had a frameshift mutation in at least one genome. A total of 52% of the frameshifts (210 sites) occurred in only one genome, but a small minority of sites were more frequently mutated. Twenty-four sites had frameshifts in at least 10 genomes, and 5 sites had frameshifts in more than 100 genomes. At least one frameshift mutation was found in each *agr* gene (Fig. 3A). We observed a total of 2,997 *agr* operons with at least one frameshift mutation, only 91 of which had two. We did not observe any *agr* operons with more than two frameshift mutations. The rate of mutations in the *agr* operon follows the expected Poisson distribution with a mean of 0.0765 (Fig. S4). The *agrC* gene had acquired the greatest number of different frameshift mutations, including truncation mutations in CC5, CC8, CC22, and CC30 (865 genomes) (Fig. 3A and B). The most frequent frameshift was the

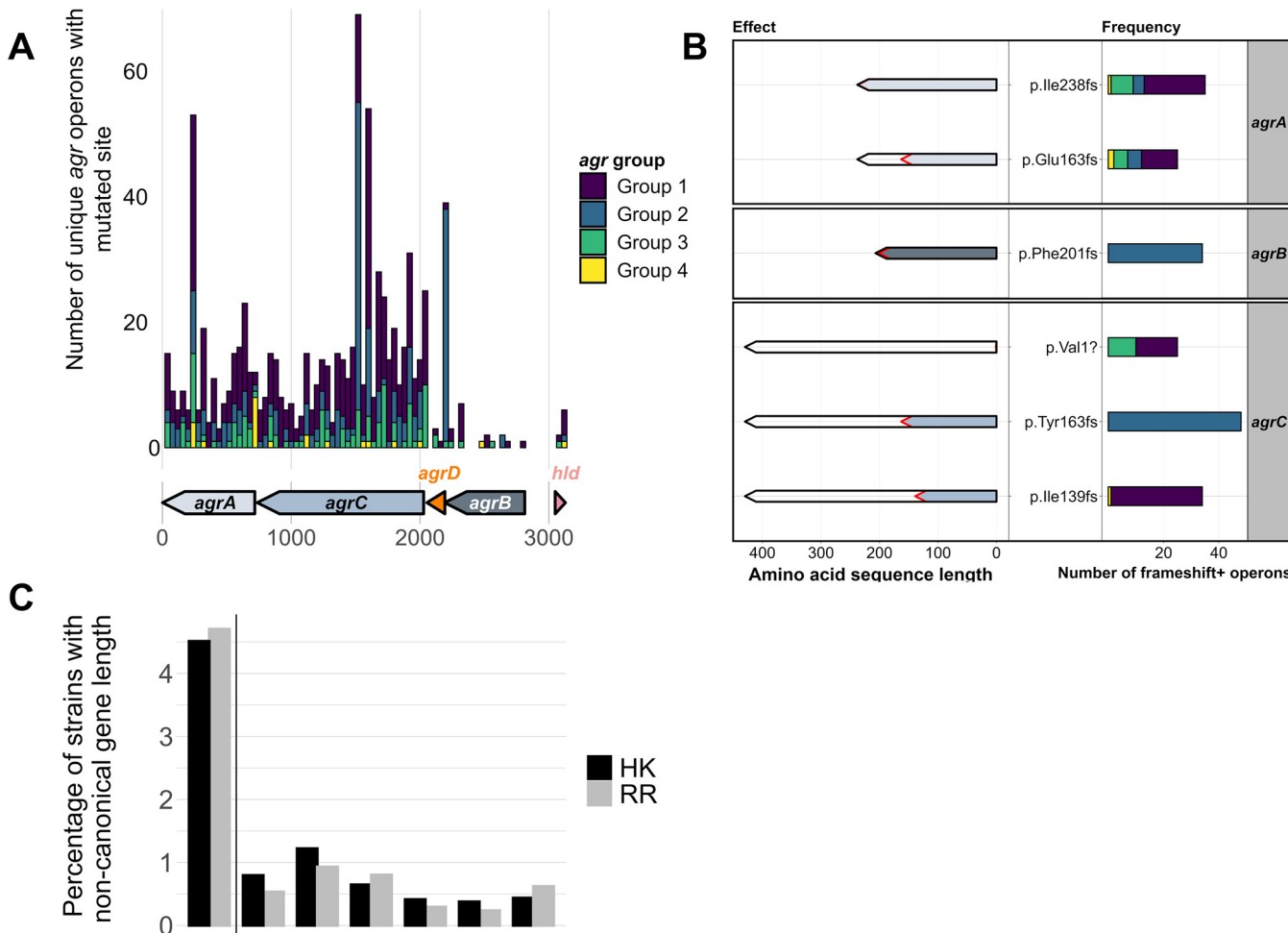

**FIG 3** Presence of putative non-functional variants of the *agr* operon. (A) Frequency of frameshift mutations in coding regions of unique *agr* operon sequences across the Staphopia database. Arrows indicate *agr* genes and bars indicate the number of frameshifts at the corresponding position (bin width = 40). Bar colors represent each *agr* group. (B) Frequency (right) and effect (left) of commonly occurring frameshift mutations across unique *agr* operon sequences. Bars are colored based on *agr* group and arrows are colored based on *agr* gene, black outlines represent canonical protein length and red outlines represent truncated protein lengths. Labels (center) indicate the amino acid change due to the frameshift mutation. (C) Normalized percentage of samples with noncanonical two-component regulator (TCS) gene lengths. Histidine kinase (HK) and response regulator (RR) genes of TCS were extracted from the Staphopia database and commonly occurring gene lengths (>5000 genomes) were excluded. The remaining strains were considered to have noncanonical gene lengths.

insertion of an adenine in the terminal end of *agrA*, occurring in 561 genomes (35 unique *agr* operons) across multiple CCs (p.Ile238fs in Fig. 3B). This mutation has previously been investigated and was found to cause delayed *agr* activation (59). Another mutation toward the end of the gene occurring in a polyA tract found in *agrB* occurred in 114 genomes (34 unique *agr* operons), mainly CC15 and CC5 (p.Phe201fs in Fig. 3B). A frameshift that resulted in the loss of the start-codon in *agrC* was observed in 86 genomes (25 unique *agr* operons) of CC8, CC30, CC22, and some other rarer CCs (p.Val1? in Fig. 3B). In relatively low frequency (69 genomes), we observed complex mutations, such as collapsed repeats, tandem duplications, and large (>30 bp) in-frame and out-of-frame indels. However, such mutations were mostly singular sporadic events, and no mutation was recurrent in more than three different genomes. Two of these cases were an insertion sequence (IS) element insertion in the *agr* operon. One was a 1,326bp insertion of an IS256 family transposon sequence found commonly in *Staphylococcus* species (60), and the other was a 1057 bp insertion of an IS1252 transposon sequence found in *Enterococcus* species (61).

Frameshifts in the delta-toxin gene (*hld*), which was present within the RNAiii transcript (Fig. 1A), were rare with only 4 genomes having one of two mutations. One was a deletion at position 76 leading to a frameshift, and one was a G to A substitution in position 44

leading to a premature stop. We found 3943 indels in RNAiii, 3931 of which were single nucleotide indels. The most common mutation occurring 2025 times was the insertion of a T at position 406 of RNAiii, which was found exclusively in *agr* group-3 genomes, suggesting that this might be a common variant. It is unknown whether these single nucleotide indels have a functional impact on RNAiii. On the other hand, we also found 8 genomes with large >30 bp indels in RNAiii which may affect function. Namely, two *agr* group-1 genomes with a 42 bp deletion at position 391, four *agr* group-2 genomes with a 41 bp insertion at position 2, and one *agr* group-3 strain with a 31 bp insertion at position 9.

We compared the number of genomes with indels in the *agr* operon to other two-component regulatory systems (TCS) to determine if the frequency of potentially deleterious mutations in the *agr* operon was significantly different. This was done by extracting the histidine kinase (HK) and response regulator (RR) genes of TCS *arlRS*, *kdpDE*, *nreBC*, *phoPR*, *srrAB*, and *walKR* from the Staphopia database and calculating the number of genomes with HK or RR genes with non-canonical gene lengths. The length of the reference gene for each HK and RR was considered the canonical length and the length of each annotated gene in the Staphopia database best matching the reference gene was identified (BLASTn). At least 42,500 hits for each HK or RR were extracted out of 42,999 genomes in the Staphopia database. Hits for each HK and RR with lengths not equal to their corresponding reference were considered non-canonical gene lengths due to indels. Frequently occurring alternate gene lengths (observed in >5000 genomes) were considered common alleles and not mutated variants. We found that the *agr* TCS had a significantly greater number of variable gene lengths compared to TCS *arl*, *kdp*, *nre*, *pho*, *srr*, and *wal* ($P < 0.0001$, negative binomial regression). When normalized to 1 kb, we found ~4.5% of all *agrC* (HK) and *agrA* (RR) genes analyzed were of variable lengths. In contrast, only ≤1.5% of all other HKs and RRs analyzed had variable gene lengths (Fig. 3C). Overall, *agrAC* had a higher percentage of non-canonical gene lengths compared to the corresponding genes from other two-component systems, suggesting higher frequencies of indels.

To estimate whether factors, such as *agr* group, clonal complex, host body site, and infection/colonization status, can serve as predictors of null mutations, we trained models using a general linear model (GLM), random forest (RF), extreme gradient boosting (XGB) and K-nearest neighbors (KNN) to predict the presence/absence of frameshift mutations in the *agr* operon. All 4 models had a high negative predictive value and low precision, which could be due to the imbalanced nature of the test data set (see Methods and Materials) (Table S2). This suggested that the likelihood of acquiring frameshift mutations in the *agr* operon could not be predicted by the site of infection and pathogenicity status alone.

**Some *agr* frameshift mutations occurred repeatedly through convergent evolution.** We noticed that certain frameshift mutations in the *agr* genes occurred frequently across different strain backgrounds. To test whether these recurrent mutations could be explained by a purely random mutational process, we simulated frameshifts in a random set of wild-type operons and compared the resulting frequency distribution to the real distribution of mutation sites in the genomes carrying *agr* frameshifts (referred to as frameshift+ genomes). We chose a set of dereplicated genomes from CC8, CC22, CC5, and CC30 to reduce sampling bias affecting frameshift counts (see Materials and Methods for dereplication strategy). These are the most abundant CCs in the Staphopia database and carry many of all identified frameshift mutations. We found that, although the total number of mutation events in the real and simulated data set was similar (~300), the number of unique sites mutated in the simulated data set was greater than the real data set. This showed that the simulated distribution was significantly different from the real distribution of frameshifts in the *agr* operon (Kolmogorov-Smirnov $P < 0.01$). (Fig. 4A). Moreover, we calculated the consistency index of each frameshift site on CC-specific maximum likelihood trees using HomoplasyFinder (62). In short, a consistency index of one for a given site on an alignment indicates that the nucleotides at that site are consistent with phylogeny, and a consistency index of 0 indicates that the nucleotides at the site are homoplasious (evolved independently of phylogeny). We found a trend of decreasing consistency index with increasing frequency of each recurring frameshift. We observed an almost identical trend

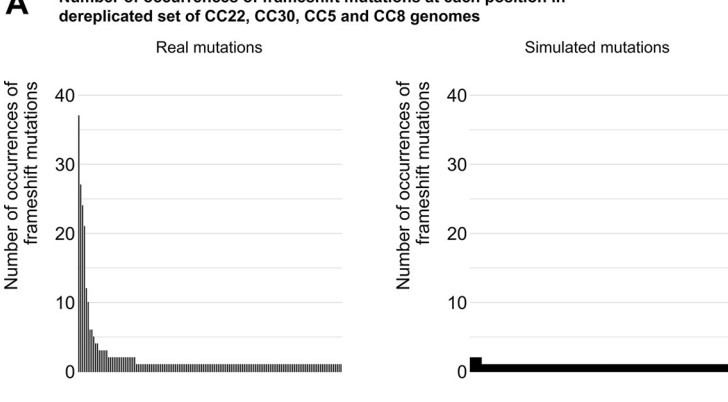

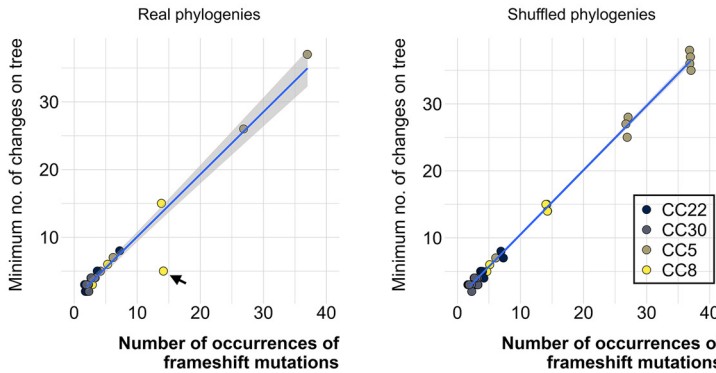

**FIG 4** Identical *agr* mutations evolve independently of phylogeny across different clonal complexes. (A) Bars show the frequency of mutations at a given site in descending order. Frequency of each mutation in a dereplicated set of CC8, CC30, CC22, and CC5 samples (left). Frequency of each mutation in a set of randomly selected CC8/CC30/CC22/CC5 *agr* operons with simulated indels (right). (B) The minimum number of changes on the tree versus the number of occurrences of frameshift mutations. Each circle represents a position on the *agr* operon that has acquired a mutation in at least 2 samples in a dereplicated set of CC8, CC30, CC22, and CC5 sequences (left). The x-axis represents the number of times the position has acquired a frameshift mutation. The consistency index and the minimum number of changes on the tree were measured at these sites for each CC from the respective phylogenetic tree (GTR+FO model, 1000 ultrafast bootstrap replicates with average bootstrap support of at least 71%) Blue line follows y = x distribution. The outlier CC8 point (black arrow) corresponds to a previously characterized *agrA* mutation that was not a true *agr* null (59). The consistency index and the minimum number of changes on the tree were measured for phylogenies for the respective CCs where the tree tips were randomly shuffled (right). One hundred shuffled trees were generated per CC.

when the tips of the trees are shuffled to have the frameshift+ genomes at random positions on the tree, mimicking non-ancestral, independent acquisition of each frameshift. The minimum number of changes on the phylogeny equalled the number of occurrences of frameshift mutations. Because these mutations were being acquired repeatedly, independent of phylogeny, they are reflected in the number of changes on the tree (Fig. 4B). The outlier mutation (Fig. 4B, black arrow) that appeared to have transmitted to multiple isolates in CC8 was an *agrA* mutation at the 3′ end found to cause delayed *agr* activation (59) and, therefore, was not a true *agr* null. Overall, this showed that there is a preference for specific sites in the *agr* operon to acquire potentially null-inducing mutations and that identical mutations can occur independently across different *S. aureus* lineages.

## DISCUSSION

In this study, we used Staphopia, the largest available database of consistently assembled and annotated *S. aureus* genome sequences (n = 42,999) to analyze patterns in the evolution and diversity of the *agr* quorum-sensing system. Our goal was to place previous work on the

evolutionary genetics of *agr* in the context of the thousands of genomes currently available. We developed a bioinformatics tool, AgrVATE, for rapid genome-based classification of *agr* specificity groups and identification of putative null mutations in the *agr* operon (https://github.com/VishnuRaghuram94/AgrVATE). Our findings to a large extent were consistent with previous studies but the increased scale of the analysis revealed new features. We confirmed that only the 4 previously known autoinducing peptides that define *agr* groups 1 to 4 were present in *S. aureus* (Fig. 1B, Fig. S3). To our knowledge, there has been no credible report of any other peptide reported, although other *Staphylococcus* species have *agr* operons encoding different cyclic peptides (63). We also found, as previously reported (10, 64), that *agrBDC* alleles are *agr* group-specific while *agrA* alleles are independent of *agr* group (Fig. 2A). In addition, we found that *agrBDC* are in linkage disequilibrium while being unlinked to *agrA* (Fig. 2D). Moreover, except for CC45, clonal complexes were exclusively linked to specific *agr* groups and specific alleles within these groups (Fig. 1C). A third major result was that, while the *agr* operon was rarely deleted completely in any strain, ~5% of all analyzed *agr* operons have at least one frameshift mutation in the coding regions, indicating that potential nonfunctional *agr* variants are relatively common (Fig. 3A and B). *agr*-defective strains have been frequently reported (16–21), and we showed through genomic comparison that *agrAC* more frequently accumulated frameshift mutations compared to other *S. aureus* two-component systems (Fig. 3C). Using the somewhat limited publicly available metadata, we could not link *agr* null strains to a particular body site or type of infection. While most of these frameshift mutations are singular evolutionary events, we found a few sites across unrelated strains that have independently acquired a disproportionately high number of frameshifts (Fig. 4A) with no evidence of long-term phylogenetic transmission (Fig. 4B), suggesting selection or a generative mechanism for high-frequency mutations. At least one of these frameshifts has been studied functionally (59), but many of these frequent frameshifts were only detected through large-scale genomic analysis reported here.

This study reemphasizes that *S. aureus* is not phylogenetically structured according to *agr* groups: i.e strains belonging to each *agr* group do not fall into their monophyletic clades (Fig. 2C) (64). This pattern can be best explained by rare homologous recombination of the *agrBCD* genes. Strikingly, all strains within a clonal complex (except the previously mentioned CC45) belonged to only a single *agr* group. While there have been reports of multiple *agr* groups within the same sequence type in *S. aureus* (54, 55), we did not observe any such instances across more than 40,000 genomes. It has been proposed that clonal complexes in *S. aureus* emerge from recombination and/or genome rearrangement events and remain stable due to the presence of barriers to recombination and HGT between CCs (65, 66). These results suggest that *agrBDC* recombination may be the impetus for the formation of clonal complexes. CC45 may be in the process of CC formation after a recent switch in *agr* group and may thus be an interesting natural laboratory for understanding the evolutionary dynamics that drive this process.

We observed strong purifying selection in *agrA*. Most of the CC-specific nucleotide changes in *agrA* were synonymous changes, with 83% of all AgrA sequences being identical. The only significant non-synonymous change we observed on a species-wide scale was a single amino-acid substitution at position 136 (9% of all AgrA sequences). This alternate AgrA (AgrA[R136]) was found only within one clade comprising CC15 and other rare CCs of *S. aureus* (Fig. 2B and C). Overall, this shows that nucleotide differences in the *agr* operons, even in operons belonging to the same *agr* group, can serve as a predictor of the subspecies and clonal complex; however, the functional impact of these alternate alleles, if any, are unknown.

We know that *agr* function is not always essential for *S. aureus* survival because nonfunctional *agr* variants are commonplace and are frequently isolated from patients (16–21). For example, there is a relatively high occurrence of nasal colonization by strains with downregulated *agr* expression in hospital settings (67). Nasal carriage is an important step for initiation of *S. aureus* infection, and it has been observed that the presence of isolates with impaired *agr* function in the bloodstream is often associated with isolates of identical *agr* function in the nasal cultures (68, 69). This suggests that complete virulence capacity is not an absolute requirement for colonization and transmission of *S. aureus* in the hospital

environment (43). However, community-associated transmission by *agr* defective strains is thought to be curtailed and the *agr⁻* strains do not remain long enough to establish a circulating population outside the initial location (45, 67). This brings to light the possible evolutionary trade-off for strains that become *agr* defective. Although some short-term transmission may have occurred, our phylogenetic analyses show no evidence of stable lineages of putative *agr* null populations (Fig. 4B). In addition, we also found that the frequency of mutations in the *agr* TCS is enriched compared to other TCS in *S. aureus* (Fig. 3C). The relatively common occurrences of independently acquired *agr* mutations suggest that they may be adaptive convergent mutations in response to specific selective pressures. It is also important to note that this study does not investigate non-synonymous substitutions and mutations in genes outside the *agr* operon which may affect *agr* activity. A recent study (70) showed that isolates with reduced toxin production need not necessarily harbor *agr* mutations. Our hemolysis results from CF *S. aureus* strains (Table S1) also support this because we saw strains without *agr* mutations showing reduced hemolysis. This highlights the multifaceted nature of *agr* mediated virulence and that the true frequency of phenotypically *agr* null is likely higher than what we report in this study.

The *Staphylococcus agr* system is a central feature of virulence gene regulation that has been studied for more than 40 years but much regarding the evolution and maintenance of *agr* remains poorly understood. There are two particularly interesting negative findings in this study: the absence of noncanonical "intermediate" AIPs in 42,999 strains, and the absence of any strain that has acquired *agrD* from *Staphylococcus* species other than *S. aureus*. The evolutionary mechanism behind the diversity of *S. aureus agr* was hypothesized to be a random mutation of the *agr* locus to give rise to multiple sequence configurations, followed by selection for only functional configurations leading to the four *agr* specificity groups that exist today (32). This model implies intermediate or transitional *agr* groups, which were presumably non-functional *agr* operons driven to extinction by diversifying selection, allowing only functional *agr* systems to become successful lineages. The absence of intermediates suggests a strong selection for maintenance for four group specificities in *S. aureus* that leaves producers of novel peptides at a disadvantage. The high number of frameshifts suggests that not producing any peptide at all may confer higher fitness in some environments and could be a viable transitory strategy. Similarly, while there is abundant evidence for HGT of antibiotic resistance genes from other *Staphylococcus* species (71–75), we did not find evidence of *S. aureus* acquiring *agr* genes encoding novel AIP specificities. Close relatives of *S. aureus*, such as *S. argenteus* and *S. schweitzeri*, share the *agr* group-1 AIP, although *S. argenteus* and *S. schweitzeri* also developed their own distinct AIPs (76). This may suggest that while a common ancestor of these three *Staphylococcus* species may have also been *agr* group-1, environmental niche selection drove the emergence of species-specific *agr* groups. It may be that AIP specificity plays a role beyond just intraspecies competition that we do not yet understand.

## MATERIALS AND METHODS

**AgrVATE workflow.** AgrVATE was written in bash and uses freely available software. AgrVATE only requires a *S. aureus* genome assembly in FASTA format as input and the outputs include the detected *agr* group, the extracted *agr* operon, and a table with any annotated variants. The installation and usage instructions as well as descriptions of all output files can be found on Github (https://github.com/VishnuRaghuram94/AgrVATE). It is recommended to run AgrVATE on Unix-based operating systems. The methods for building AgrVATE and running it on Staphopia genomes are outlined below.

**(i) Identifying a unique set of 31mers for each *agr* group.** We first assigned *agr* groups based on the AIP amino acid sequence to all genomes in the Staphopia database (46) where a canonical AgrD protein was annotated by Prokka v1.14.6 (40,812 AgrDs) (77). We then extracted the *agr* operons from these 40,812 genomes by *in silico* PCR using usearch v11.0.667_i86linux32 (51). To identify kmers unique to each *agr* group, DREME v5.1.1 (78) was used to identify 31 bp kmers (31mers) that were unique to the *agr* operon of each *agr* group, resulting in four distinct groups of 31mers (E value < 0.0001). AgrVATE uses this output of 31mers unique to each *agr* group as a database to conduct a BLASTn v2.10.1 (79) search against an assembly of a given *S. aureus* genome to identify the *agr* group. We also used AgrVATE to reassign *agr* groups to the preliminary set of 40,812 genomes and the group assignments matched in 40,725 cases. In the 87 cases where the initially assigned *agr* group did not match the AgrVATE assignment, we found that the genome assemblies were contaminated with another *S. aureus* isolate of a different *agr* group. In these cases, AgrVATE will assign the *agr* group with the most kmer matches while also noting that more than one group was found.

**(ii) *Agr* operon and *agr* gene extraction.** *In silico* PCR was performed for 42,999 *S. aureus* whole-genome sequences in the Staphopia database using the usearch -search_pcr tool (51) with the following primers:

5'AAAAAAGGCCGCGAGCTTGGGAGGGGCTCA'3 and 5'TTATATTTTTTTAACGTTTCTCACCGATGC'3. Both primers were required to bind, and 8 mismatches in total were allowed. Extracted *agr* operons were clustered with 100% identity using usearch -fastx_uniques (51) to obtain all possible unique *agr* operon configurations. This unique set of operons was annotated using Prokka v1.14.6 (77), *agr* genes were extracted and clustered again with the same parameters to obtain all possible nucleotide and amino-acid configurations of each *agr* gene.

**(iii) Identifying variants in the *agr* operon.** The most frequently occurring *agr* operon nucleotide configuration was determined for each *agr* group and used as a reference. Variant calling was performed using snippy v4.6.0 (52). Only the loss of start, gain of stop, and frameshift mutations occurring within the coding regions of the *agr* operon were considered possible non-functional variants. AgrVATE filters the snippy output and reports the above-mentioned mutations in tabulated format.

**Staphopia metadata.** Metadata associated with *S. aureus* genomes submitted to the NCBI Short Read Archive was downloaded as a table using the Run Browser tool. This was then subjected to a series of bioinformatic filters to clean up key fields, such as collection date and location, host body site, and host status. Additional data from supplemental tables of several published *S. aureus* genome sequencing studies were also added. The data and scripts can be accessed at https://github.com/Read-Lab-Confederation/staphopia _metadata/. The table used in this study was 'Stage3.4.csv' (commit 4548f17, 2020-08-14).

**Whole-genome phylogeny and linkage disequilibrium.** The Staphopia database non-redundant diversity (NRD) set which contains 380 genomes was filtered to contain only genomes where the full *agr* operon was extracted by AgrVATE and all four *agr* genes as well as the genes up and downstream of the *agr* operon were annotated by Prokka v1.14.6 (77), resulting in 355 genomes. These 355 genomes were further filtered to include only genomes where the *agr* group prediction was unambiguous, leading to 334 genomes. A core genome alignment was constructed for these 334 using parsnp v1.5.3 (80) and this alignment was used to build a maximum likelihood phylogeny using IQ-TREE v1.6.12 (57) with the GTR+FO model and 1000 ultrafast bootstrap replicates using *S. argenteus* as the outgroup (GenBank accession number AP018562.1). The outgroup was then removed, and the tree was reconstructed with the tip closest to the outgroup (ST93) as the root. The resulting tree was then plotted using the R package ggtree (81).

The genomic region comprising the *agr* operon and 1000 bp on each side was extracted from the initial filtered NRD set of 355 genomes and this region was aligned using snippy-core v4.6.0 (52). The full alignment (core.full.aln) file was then converted to a VCF file using snp-sites v2.5.1 (82), and this VCF file was used to calculate Pearson's coefficient for linkage disequilibrium (LD) using plink v1.90b6.21 (options: –r2 inter-chr) (58). The resulting table was used to build an LD plot using R.

**Comparing indel rate of *agr* to other *S. aureus* global regulators.** USA 300 strain NRS384 (accession number NZ_CP027476.1) was used as a reference to extract histidine kinase (HK) and response regulator (RR) genes belonging to different *S. aureus* two-component regulatory systems (*arlRS*, *kdpDE*, *nreBC*, *phoPR*, *srrAB*, and *walKR*). A TCSs was chosen if the number of gene ontology enrichment hits exceeded 10 in the regulon of a constitutive RR strain where all other TCSs are deleted (83). The length of the reference gene for each HK and RR was considered the canonical length and the length of each annotated gene in the Staphopia database best matching the reference gene was identified (BLASTn). Hits for each HK and RR with lengths not equal to their corresponding reference were considered noncanonical gene lengths due to indels. Commonly occurring variant gene lengths (>5000 strains) were still considered canonical and filtered out. We performed negative binomial regression on 1 kb normalized count data of frequency of variable gene lengths across the TCS offsetting for canonical gene length and a total number of genes. Tukey's method was used for multiple comparisons. Statistical tests were performed using the *nb.glm* function from the MASS R package (84) and multiple comparisons were performed using the emmeans R package (85).

**Classifiers for predicting frameshift mutations in the *agr* operon.** R package caret (86) was used to train classifiers using repeated k-fold cross-validation (10-fold, 3 repeats). The training data set comprised 400 randomly sampled frameshift-positive and frameshift-negative strains each to overcome the imbalanced representation of each class (24:1). Strain metadata was obtained from the Staphopia database and only features annotated in > 25% of all strains were included. Strains with unknown host status and host body site data were filtered out. The final data set contained 11500 frameshift-negative and 486 frameshift positive strains.

**Dereplication of Staphopia database genomes.** CC5, CC8, CC22, and CC30 genomes were separated into frameshift+ and wild-type groups within each CC based on the presence/absence of frameshift mutations in the *agr* operon. For each CC, the two groups were independently clustered using a Mash (56) distance threshold of 0.0005 and a representative for each cluster was chosen at random. This Mash distance threshold was chosen empirically based on a comparison of pairwise Mash distances and pairwise SNP distances within the Staphopia database NRD set. Mash distances <0.0005 represent a median distance of 47 with a maximum of 282 (Fig. S5). SNP distances were calculated from parsnp v1.5.3 (80) core genome alignments using snp-dists v0.7.0 (87). Each frameshift+ cluster was limited to a size of 50 genomes and each wild-type cluster was limited to a size of 200 genomes to produce evenly sized clusters and to prevent underrepresentation of frameshift+ genomes. In total, 1093 genomes represented CC5, 1110 CC8, 404 CC22, and 705 CC30.

**Simulating mutant *agr* operons.** A combined total of 312 genomes from the dereplicated set of CC5, CC8, CC22, and CC30 genomes were frameshift+, which equated to 312 mutational events as each genome in the dereplicated set contained only one frameshift in the *agr* operon. To simulate a similar number of mutations, we used Mutation-Simulator v2.0.3 (https://github.com/mkpython3/Mutation-Simulator) to induce insertions or deletions at a rate of 0.0002 (parameters: –insert 0.0002 –deletion 0.0002) in 350 randomly selected wild-type genomes from the dereplicated set, leading to 307 simulated indels.

**Calculating consistency indices.** Core genome alignments of the dereplicated genomes for each CC were performed using parsnp v1.5.3 (80) and maximum likelihood phylogenetic trees were constructed using IQ-TREE v1.6.12 (57) using the GTR+FO model with 1000 ultrafast bootstrap replicates. Java version of HomoplasyFinder was fed the phylogeny and a presence-absence matrix of *agr* operon frameshift positions to

obtain the consistency index for each position. The phylogenies for each CC were then imported to R using the ggtree package (81) and the tip labels were randomized 100× to produce 100 shuffled trees. The consistency indices for *agr* operon frameshift positions were calculated for all shuffled trees in the same fashion. Kolmogorov-Smirnov test was used to compare the distributions of consistency indices using the R function *ks.test*.

**Sputum sample collection and whole-genome sequencing.** The whole-genome sequences for 64 out of the 91 CF strains analyzed in this study are associated with a previous publication and can be found in the accession number PRJNA480016 (88). The remaining 27 strains are from sputum samples provided by 3 CF patients and can be found in accession number PRJNA742745. The methods for processing these 24 strains are as follows.

Sputum samples were collected from patients at the Emory Adult Cystic Fibrosis Center and spread onto mannitol salt agar (MSA) the same day. Both volumes of 10 $\mu$L or 100 $\mu$L of resuspended sputum were plated for each sample. Three sputum samples from three different patients were collected and processed as mentioned above in the laboratory of Stephen P. Diggle at Georgia Institute of Technology. From each sample, 4 to 8 single colony isolates were picked, grown overnight in Luria Broth media, and restreaked on *Staphylococcus* isolation agar (SIA) for further purification before being frozen with 25% glycerol and stored at −80°C. SIA agar is composed of 30 g/L trypticase soy broth, 15 g/L agar, and 70 g/L NaCl. At least one 'pool' or population sample was collected per patient by scraping all remaining colonies on a single inoculation loop, resuspending the collected colonies in Luria Broth, and incubating the liquid culture overnight at 37°C. Overnight cultures of population samples were also further purified on SIA before being made into frozen stocks. Population samples are always recovered by scraping the entire plate, never as single colonies, throughout the rest of the experiments. Hemolysis phenotyping was conducted for all single colony isolates and population samples using Congo-Red agar as previously described (88).

To extract genomic DNA for sequencing, each sample was streaked on SIA. An inoculation loop was used to collect cells directly from the plate and one loop-full of cells was suspended in 50 mM EDTA. To lyse the cells, 20 $\mu$L of freshly prepared 10 mg/mL lysozyme and 100 $\mu$L of 5 mg/mL lysostaphin were added to the cell mixtures which were then incubated for 1 h at 37°C. Genomic DNA was then extracted using the Promega Wizard Genomic DNA purification kit. All samples were sequenced at the Microbial Genome Sequencing Center (Pittsburgh, PA, USA) using the Illumina Nextera kit on the NextSeq 550 platform. Single colony isolates were sequenced at a depth of 150 Mb and population samples were sequenced at a depth of 625 Mb. Raw paired-end sequence files were screened for quality and minimum length using FastQC v0.11.9 (89). Raw sequence files were then fed into the Bactopia analysis pipeline version 1.4.10 (90). Bactopia output was used to determine sequence type and clonal complex identities for each sample. Assemblies produced by Bactopia were then analyzed in AgrVATE for *agr* type and frameshift status.

**Data availability.** Source code for AgrVATE as well as the R code, supplemental information and data sets for generating the figures in this study can be found at https://github.com/VishnuRaghuram94/AgrVATE. CF isolate genome sequences used in this study can be found under BioProject accessions PRJNA480016 and PRJNA742745. The accessions, *agr* groups, sequence type, and clonal complex, and frameshift information for 40,890 *S. aureus* genomes used in this study can be found in Data Set S1.

## SUPPLEMENTAL MATERIAL

Supplemental material is available online only.
**SUPPLEMENTAL FILE 1**, XLSX file, 2.5 MB.
**SUPPLEMENTAL FILE 2**, PDF file, 0.4 MB.

## ACKNOWLEDGMENTS

Bacterial isolates and human subject samples were provided by the CF Biospecimen Repository at the Children's Healthcare of Atlanta and Emory University CF Discovery Core. We thank the Diggle Lab at the Georgia Institute of Technology for obtaining the Cystic Fibrosis sputum samples and isolating the *S. aureus* populations. We also thank Rachel Done, Dina Moustafa, Justin Luu, Cristian Crisan, Katrina Hofstetter, and Brooke Talbot for providing constructive comments on the manuscript.

V.R. and T.D.R. were supported by grant number AI139188 from the National Institutes of Health (NIH). A.M.A. was supported by the Infectious Disease Across Scales Training Program at Emory University. This work was also supported by grants from the Cystic Fibrosis Foundation (GOLDBE19P0) and the NIH (R21AI48847) to J.B.G.

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
