## [Reviewer comments · Microbiology Spectrum]

Microbiology Spectrum

Species-wide phylogenomics of the *Staphylococcus aureus agr* operon reveals convergent evolution of frameshift mutations

Vishnu Raghuram, Ashley Alexander, Hui Qi Loo, Robert Petit, Joanna Goldberg, and Timothy Read

Corresponding Author(s): Timothy Read, Emory University School of Medicine

Review Timeline:

Submission Date:	August 23, 2021
Editorial Decision:	October 13, 2021
Revision Received:	November 3, 2021
Editorial Decision:	November 29, 2021
Revision Received:	December 2, 2021
Accepted:	January 3, 2022

Editor: Gaurav Sharma

Reviewer(s): Disclosure of reviewer identity is with reference to reviewer comments included in decision letter(s). The following individuals involved in review of your submission have agreed to reveal their identity: Nur A Hasan (Reviewer #1)

Transaction Report:

DOI: <https://doi.org/10.1128/spectrum.01334-21>

October 13, 2021

Dr. Timothy D Read
Emory University School of Medicine
Atlanta

Re: Spectrum01334-21 (Species-wide phylogenomics of the *Staphylococcus aureus agr* operon reveals convergent evolution of frameshift mutations)

Dear Dr. Timothy D Read:

Thank you for submitting your manuscript to Microbiology Spectrum. This is a well-executed project that provides several useful insights for the scientific community. Now we have received the comments from both reviewers for your submitted manuscript. The first reviewer has suggested minor revisions, whereas the second reviewer has recommended additional experiments and explanations. While both expert reviewers have suggested different scopes for the revision, I conservatively side with the opinion that suggests more effort. The comments from Reviewer 2 are the most critical ones, therefore, please give more emphasis on those comments. Altogether, please address all the comments by both reviewers. Please ensure that the added comments are well explained throughout the text and supported by clearly described methods.

When submitting the revised version of your paper, please provide (1) point-by-point responses to the issues raised by the reviewers as file type "Response to Reviewers," not in your cover letter, and (2) a PDF file that indicates the changes from the original submission (by highlighting or underlining the changes) as file type "Marked Up Manuscript - For Review Only". Please use this link to submit your revised manuscript - we strongly recommend that you submit your paper within the next 60 days or reach out to me. The detailed information on submitting your revised paper is provided below.

Link Not Available

Sincerely,

Gaurav Sharma, Ph.D.

Editor, Microbiology Spectrum
<https://sites.google.com/view/sharmaglab/>

Journals Department
Reviewer comments:

Reviewer #1 (Comments for the Author):

The manuscript by Raghuram et. al. presents a species-wide phylogenomic analysis of *agr* operon in *S. aureus*. They analyzed over 40,000 genomes from the Staphopia database and report that ~5% of the genomes include frameshift mutation in the *agr* operon. However, the phylogenomic analysis suggested that the evolutionary path of strains with *agr* frameshift mutation leads to a dead end. Furthermore, the authors developed new software for *Agr* genotyping in *S. aureus*. Overall, this is an excellent study, very well designed, and nicely written. It adds new knowledge and capability in understanding *S. aureus* virulence. Below please find some minor comments:

1. In the introduction, the authors should discuss the *Agr* phase variability as described by Ghor et al., 2019.

2. Please consider replacing "Staphopia" with "Staphopia database" throughout the manuscript.
 3. Figure 2 has poor resolution, please improve it. Additionally, 2C needs to be bigger than what it is now.
 4. I did not see much utility of Figure 3 as the main figure, hence suggest moving it to supplementary figures.
-

Reviewer #2 (Comments for the Author):

This manuscript reports a major effort to characterize an enormous collection of *S. aureus* genomes for truncation mutations in the *agr* locus. Such mutations are known to emerge frequently during colonization with *S. aureus*. As such, further understanding their emergence and distribution is an important question. While I admire the authors for choosing this interesting direction, more work is needed to reach the conclusions listed in the abstract.

Major

-A major claim in the abstract is that "Phylogenetic patterns suggested that strains with *agr* frameshifts were evolutionary dead ends." Similarly, the discussion mentions that the same mutations were "independently acquired." This is an important claim that impacts how other data presented here are interpreted. Either support or contrast with the result would be interesting if characterized more convincingly: the extent to which *agr* null mutants persist and spread between individuals is currently unknown, as is the likelihood of the same mutation appearing independently. However, this result is poorly supported by the presented data and appears to be in fact contradictory to the data provided. The outlier point in 4B suggests a mutation that occurred early on the phylogeny that is now in many isolates. Moreover, the distribution of specific mutations across particular *agr* types as shown in Figure 3B also suggests identity-by-descent. Lastly, it is unclear how the dereplication was done for the analysis in 4B impacts results; it appears to me that this step would overestimate 'dead ends' and underestimate the spread of *agr* null alleles. An analysis that counts the number of independent occurrences on the tree from the full data set is required. If there is a limit to how many sequences can be imported, complete smaller subsets of the phylogeny should be used.

-Relatedly, the authors should consider reporting the "minimum number of mutations on tree" rather than the consistency index. This would be easier to interpret, as we the claim that all mutations are dead ends would produce an expected $y=x$ line on a plot of # of isolates vs # of mutations. This can be calculated from the reported consistency index (<https://github.com/JosephCrispell/homoplasyFinder/wiki/Calculating-consistency-index>)

-Are multiple frameshift mutations in *agr* and/or indel mutations in RNAiii found in the same genomes? Or is there a signal for overdispersion (i.e. more cases of 1 mutation per pathway per genome than expected if you reshuffle these mutations)?

-Line 292: I would have expected a comparison of # unique frameshift mutations per bp or per gene across these genes, rather than analysis of gene length distributions. Are all genes in this operon equally likely to acquire mutations, condition by their gene length?

Minor:

Figure 4B: I think the x-axis should be labeled "number of isolates". If this is accurate, a clearer label should be used. If not, what is meant by an 'occurrence' should be more clearly explained.

The manuscript should make it easier to align data from different panels by indicating the *agr* types of various clonal complexes.

Line 291- I'm not sure I understand this: "Commonly occurring variant gene lengths (>5000 genomes) were still considered canonical and filtered out." Perhaps a typo?

-Line 188: This statement needs a P-value.

-Line 210: ANI or other metrics of diversity that account for gene length would be helpful here.

Line 215: Could this be the identity by descent, given how small this gene is? Perhaps the private alleles in CC15 and CC5 are both derived from this shared allele?

Line 222: How different are these *agr*BDC alleles? 1 mutation away? This is a similar concern as the above.

How is the tree in Figure 2C rooted? The variation root-to-tip distance looks suspicious.

-Line 363 is worded too strongly. There is clearly some phylogenetic signal, as indicated by the clonal complex level analysis.

Minor text:

-Line 157: An exact number would be appreciated here rather than forcing the reader to the SI.

-Line 178: The comparison between patients and isolates is misleading. An isolate and a population should not be expected to behave similarly in terms of genetic diversity.

-Line 237: This is misleading, the wording suggests a monophyletic trait. This trait is not monophyletic as shown in 2C.

Staff Comments:

Preparing Revision Guidelines

Please return the manuscript within 60 days; if you cannot complete the modification within this time period, please contact me. If you do not wish to modify the manuscript and prefer to submit it to another journal, please notify me of your decision immediately so that the manuscript may be formally withdrawn from consideration by Microbiology Spectrum.

**RESPONSE TO REVIEWERS**

**Species-wide phylogenomics of the *Staphylococcus aureus agr* operon reveals convergent**
**evolution of frameshift mutations**

Major changes in this version are: a) new Supplemental Figure S4 with new analysis showing
that frameshift mutations in the dataset follow a Poisson distribution, implying a random process,
b) new Supplemental Figure S5 with new analysis of SNP distances between strains with mash
distance of < 0.0005 , c) changes to Figure 3C that show more clearly now that *agr* is unusually
rich in indels among two-component systems in *S. aureus*. d) Changes to fig 2C to include the
clonal complexes in the phylogeny to better illustrate the point that *S. aureus* is not
phylogenetically structured by groups. We also made minor changes to figure 4B and made
numerous smaller clarifications to the text and the figures some of which were not directly asked
for by the reviewers.

Our responses to the reviewers are in red. The line numbers specified in the reviewer's
comments correspond to the first version of the manuscript while the line numbers we have
specified in our responses correspond to the revised version.

**Reviewer 1**

1. In the introduction, the authors should discuss the Agr phase variability as described by
Ghor et al., 2019.

*Added sentence to line 93 and added reference (41). The index for all citations post 41*
*now are increased by 1.*

2. Please consider replacing "Staphopia" with "Staphopia database" throughout the
manuscript.

*Done. (Lines 174, 175, 201, 205, 223, 238, 292, 294, 296, 323, 455, 469, 492, 516, 529,*
*533, 991, 995, 1017, 1026, 1066)*

3. Figure 2 has poor resolution, please improve it. Additionally, 2C needs to be bigger than
what it is now.

*Done*

4. I did not see much utility of Figure 3 as the main figure, hence suggest moving it to
supplementary figures.

*Figure 3 shows that the same mutation on the agr operon can occur across different agr*
*groups, as well as shows that the rate of mutations in the agr operon is higher than other*
*TCSs in staph, both important conclusions for the paper. Therefore, we believe,*
*especially in the light of there only being 4 figures, Fig3 has a place in the main text.*

**Reviewer 2**

**Major**

1. A major claim in the abstract is that "Phylogenetic patterns suggested that strains with
*agr* frameshifts were evolutionary dead ends." Similarly, the discussion mentions that
the same mutations were "independently acquired." This is an important claim that
impacts how other data presented here are interpreted. Either support or contrast with
the result would be interesting if characterized more convincingly: the extent to which
*agr* null mutants persist and spread between individuals is currently unknown, as is
the likelihood of the same mutation appearing independently. However, this result is
poorly supported by the presented data and appears to be in fact contradictory to the
data provided.

In response to R2's thoughtful points we have made revisions that further back our
conclusions that the *agr* frameshift null mutations were acquired independently in
multiple lineages. We think that the cause of the disagreement may be one of degree.
We do not believe that *agr* null strains are completely unable to transmit (and have
cited literature in the first version that gives evidence for this). Instead, we believe
that transmission is suppressed, which leads to *agr* nulls being removed within a few
transmission generations.

a. The outlier point in 4B suggests a mutation that occurred early on the phylogeny
that is now in many isolates.

This particular mutation is the *agrA* mutation already characterized by Novick
[ref 59] as not "true" phenotypically *agr* null and was mentioned in our original
text (line 264). Therefore, we expect this mutation to be maintained vertically. We
have now highlighted the outlier in Fig 4B, included a comment in the results
section (line 339 – 341) and in the figure legend (line 1041).

b. Moreover, the distribution of specific mutations across particular agr types as
shown in Figure 3B also suggests identity-by-descent.

As shown in Fig 2C, the same *agr* group can belong to different clonal complexes
and these clonal complexes don't have to be closely related. As shown in fig 4B,
these repetitive mutations occur in different clonal complexes though some of
them are of the same *agr* group. And as shown in 3B itself, the same mutation can
occur in different *agr* groups too, meaning they evolved independently in multiple
*agr* groups. Collectively, the data do not suggest identity by descent.

c. Lastly, it is unclear how the dereplication was done for the analysis in 4B impacts
results; it appears to me that this step would overestimate 'dead ends' and
underestimate the spread of agr null alleles.

As we originally stated in line 398 to line 408, we do agree that defective *agr* is
not an absolute barrier to transmission, and we noted within-hospital transmission
of *agr*- strains have been documented (ref 43). Our dereplication strategy
collapses closely related strains into clusters, all of which have a Mash estimated
genomic distance < 0.0005. We have added a supplemental figure (Fig S5) that
shows this mash distance corresponds to a left-weighted SNP distribution with a
median distance between strains of 47 and a maximum of 282. The corresponding
methods have been added to the "Dereplication of Staphopia database genomes"
methods section (line 536 – 540). These SNP distances would translate to an
approximate average of 1-2 transmission events, with a maximum of 8 (Hall *et al*,
2019, doi: [10.7554/eLife.46402](https://doi.org/10.7554/eLife.46402)). With this dereplication rationale in mind, our
results show no evidence of an *agr*- strain crossing the close-relatedness

threshold, suggesting that though they may transmit short-term, the strains do not
become established populations circulating in communities. We have made this
point clearer in our discussion (line 408)

88 d. An analysis that counts the number of independent occurrences on the tree from
89 the full data set is required. If there is a limit to how many sequences can be
imported, complete smaller subsets of the phylogeny should be used.

To offset the redundancy in *S. aureus* genomes as well as to reduce the
oversampling of specific clonal complexes in NCBI, we opted to construct
phylogenies from dereplicated datasets. This approach reduces the ascertainment
bias in our data which would otherwise be present if we constructed a
phylogenetic tree from > 40,000 genomes.

2. Relatedly, the authors should consider reporting the "minimum number of mutations
on tree" rather than the consistency index. This would be easier to interpret, as we the
claim that all mutations are dead ends would produce an expected $y=x$ line on a plot
of # of isolates vs # of mutations. This can be calculated from the reported
consistency index

([https://github.com/JosephCrispell/homoplasyFinder/wiki/Calculating-consistency-](https://github.com/JosephCrispell/homoplasyFinder/wiki/Calculating-consistency-index)
[index](https://github.com/JosephCrispell/homoplasyFinder/wiki/Calculating-consistency-index))

We changed figure 4B accordingly to show minimum no. of mutations on the tree as
the y axis and as expected it does produce a $y=x$ line. We have also reflected this
change in the results section (line 336 – 339) and in the figure legend for 4B.

3. -Are multiple frameshift mutations in *agr* and/or indel mutations in RNAiii found in
the same genomes? Or is there a signal for overdispersion (i.e. more cases of 1
mutation per pathway per genome than expected if you reshuffle these mutations)?

We have included the exact number of *agr* operons with frameshift mutations (Line
257 – 260). We only observed a maximum of 2 frameshifts per operon which were
also extremely rare. We included a supplemental figure (Fig S4) that shows the rate
of mutations observed in the *agr* operon compared to the expected rate as modelled
by a Poisson distribution. Our data follow the expected distribution.

4. Line 292: I would have expected a comparison of # unique frameshift mutations per
115 bp or per gene across these genes, rather than analysis of gene length distributions.
Are all genes in this operon equally likely to acquire mutations, condition by their
gene length?

We normalized our variable gene-length counts to 1kb and we observe that while
*agrA* and *agrC* are ~equally likely to acquire mutations, still maintain higher
likelihood of mutations than other TCSs. We have updated Fig 3C and the results
section (line 302 – 304) accordingly. We accept that reporting gene length
distributions is an approximation. However, we felt reporting unique frameshifts for
12 additional genes across 40,000+ strains is beyond the scope of this one figure.

**Minor**

1. Figure 4B: I think the x-axis should be labeled "number of isolates". If this is
accurate, a clearer label should be used. If not, what is meant by an 'occurrence'
should be more clearly explained.

We have changed the x axis labels of Fig 4B to “number of occurrences of frameshift
mutations”. each circle corresponds to a particular position on the *agr* operon that has
acquired a frameshift mutation in a given clonal complex. The x axis shows the
number of times that particular position has acquired a frameshift mutation across a
particular clonal complex. We have clarified this in the figure legend (line 1037).

2. The manuscript should make it easier to align data from different panels by indicating
the *agr* types of various clonal complexes.

Fig 1C indicates the *agr* groups of various clonal complexes. We also modified figure
2C to include this information.

3. Line 291- I'm not sure I understand this: "Commonly occurring variant gene lengths
(>5000 genomes) were still considered canonical and filtered out." Perhaps a typo?

If the TCS gene length was different from the reference, and we observed that exact
gene length for that gene across more than 5000 genomes, we did not count them as
variants and considered them commonly occurring alleles. Much like how the
*agrBDC* alleles have different gene lengths based on the *agr* group. We changed this
sentence to make it clearer (line 298)

4. -Line 188: This statement needs a P-value.

Performed a chi-sq test and included the p value. (line 189)

5. Line 210: ANI or other metrics of diversity that account for gene length would be
helpful here.

We clustered individual *agr* genes independently of each other and only grouped
100% identical nucleotide sequences into the same cluster as stated in line 212. As we
are not comparing the diversity between *agr* genes, we did not account for gene

length. Based on the cluster sizes, longer genes like *agrC* and *agrA* appear to have
increased diversity compared to shorter genes like *agrB* and *agrD*.

6. Line 215: Could this be the identity by descent, given how small this gene is? Perhaps
the private alleles in CC15 and CC5 are both derived from this shared allele?

While that is a possibility, we believe the more parsimonious explanation is that it is
the result of a recent recombination event rather than a vestigial ancestral allele and
numerous implied deletion events.

7. Line 222: How different are these *agrBDC* alleles? 1 mutation away? This is a similar
concern as the above.

The average within-*agr* group SNP distance is 15 and between-*agr* group SNP
distance is 167 (included in line 215 – 216). Just like individual *agr* genes, we also
clustered the operon based on 100% nucleotide sequence identity. For CC45
specifically, the average SNP distance between *gp1* and *gp4* alleles was 179 (min
175, max 185). We included this in line 226.

8. How is the tree in Figure 2C rooted? The variation root-to-tip distance looks
suspicious.

The inferred tree was unrooted. We revised Fig 2C by rooting it using an ST93 strain,
which is the strain closest to the outgroup if the tree was rooted with *S. argenteus*. We
have added the rooting strategy to the methods section (line 498 – line 502)

9. Line 363 is worded too strongly. There is clearly some phylogenetic signal, as
indicated by the clonal complex level analysis.

We modified fig 3C to include the clonal complexes. While the tree is structured by
CC, the CCs are not structured by *agr* groups, i.e the tree doesn't neatly split into 4

clades each corresponding to a particular *agr* group. Therefore, we believe that *S.*
*aureus* is not phylogenetically structured by the *agr* group (line 376).

10. Line 157: An exact number would be appreciated here rather than forcing the reader
to the SI.

Done. Line 159

11. Line 178: The comparison between patients and isolates is misleading. An isolate and
a population should not be expected to behave similarly in terms of genetic diversity.

We removed the statement that makes the comparison between patients and isolates –
line 180.

12. Line 237: This is misleading, the wording suggests a monophyletic trait. This trait is
not monophyletic as shown in 2C.

We clarified the statement to say it is limited to one clade (line 240 – 241). We
removed the word “monophyletic” from line 393.

**Misc. changes not mentioned in reviewer comments:**

1. Added missing period in line 229

2. Fixed typo in line 378 – changed Fig3C to Fig2C

3. Added reference to Fig 3C in line 411

4. Removed the word “Although” from line 431

5. We filtered our dataset for the phylogenetic tree in Fig 2C to exclude genomes with
ambiguous *agr* group calls, bringing the total number of genomes used down from 355 to
334. We reflected this change in the results (line 238) and the “Whole genome phylogeny
and Linkage Disequilibrium” methods section (Line 495 – 497, line 504).

6. Added names “Cristian Crisan” and “Katrina Hofstetter” to acknowledgements section.

November 29, 2021

Dr. Timothy D Read
Emory University School of Medicine
Atlanta

Re: Spectrum01334-21R1 (Species-wide phylogenomics of the *Staphylococcus aureus agr* operon reveals convergent evolution of frameshift mutations)

Dear Dr. Timothy D Read,

Thank you for submitting the revised version of your manuscript to Microbiology Spectrum. Now we have received the comments from both reviewers for your submitted manuscript. The first reviewer has no comments, however, the second reviewer still has additional comments. Altogether, I am suggesting a minor revision for this version. Please ensure that the added comments are well explained throughout the text and supported by clearly described methods.

Link Not Available

Sincerely,

Gaurav Sharma
<https://sites.google.com/view/sharmaglab/>
Editor, Microbiology Spectrum

Journals Department
Reviewer comments:

Reviewer #1 (Comments for the Author):

I am satisfied with the revisions.

Reviewer #2 (Comments for the Author):

While the authors make compelling descriptions of their findings in the response-to-reviewer (R2R) document, the abstract remains problematically muddled and overclaiming, which can be fixed with just changes to the text. For the two claims I mention below, care should be taken throughout to avoid overclaiming in other places of the text-but I've highlighted the abstract in detail to explain the problem. I also have a few remaining minor concerns.

1)

The abstract states " More than five percent of genomes were found to have frameshift mutations in the agr operon. Though most mutations occur only once in the entire species, we observed a small number of recurring mutations evolving convergently across different clonal lineages. Phylogenetic patterns suggested that strains with agr frameshifts were evolutionary dead ends." Together, these statements make the reader think that the authors will show evidence that: (A) that agr mutants cannot transmit to another person; and (B) Any incidence of finding 2 genomes (after genome-wide dereplication) from the same agr type emerges from convergence, rather than identity by descent or recombination.

In the R2R document, the authors claim they are not intending to claim A. However, the use of the phrase 'dead end' in the abstract does have strong connotations for many readers. A quantitative or relative description of how widespread these mutations are should be used instead.

In the R2R document, the authors claim they are not intending to claim B. They rightfully note exceptions in which identity-by-descent (IBD) is likely and mention that the point is a matter of degree. However, the abstract explicitly leaves out the notion of IBD and a naïve reader would think the authors think this third possibility is to be ignored.

Both issues could easily be fixed with a simple rewriting. For example, the authors could write "More than five percent of genomes were found to have frameshift mutations in the agr operon. XX of these frameshifts were due to unique mutations found only once, suggesting that *S. aureus* with agr frameshifts are less fit in the long term. Despite this, we observe cases where the same mutation emerges in distinct clonal lineages, reinforcing the adaptive nature of these mutations in the short term"

2) Figure 4A is now more surprising that 'occurrence' has been better defined (though this should be more explicitly defined in the figure itself -- e.g. '# of CC w same mutation'). The word occurrence, if I am understanding correctly, refers to the number of CCs in which a mutation is found. The convergence here is remarkable and suggestions recombination might be an explanation. If so, it would be useful to color these by agr gene (agrA may recombine more across CC complexes). Alternatively, if 'occurrence' means 'unique agr operon' (a phrase used earlier in the text), this is problematic as an operon can become unique after the acquisition of an agr frameshift mutation due to other mutations later occurring on that background.

3) It is not completely clear to me why the authors have discounted the possibility of recombination of a nonfunctional agr allele as a cause for its occurrence in multiple CCs.

4) A supplemental table of every genome analyzed, its CC, agrA type, agr group, and any frameshifts noted would be a useful tool for the field.

Staff Comments:

Preparing Revision Guidelines

Please return the manuscript within 60 days; if you cannot complete the modification within this time period, please contact me. If you do not wish to modify the manuscript and prefer to submit it to another journal, please notify me of your decision immediately so that the manuscript may be formally withdrawn from consideration by Microbiology Spectrum.

RESPONSE TO REVIEWERS

Species-wide phylogenomics of the *Staphylococcus aureus agr* operon reveals convergent evolution of frameshift mutations

In this revision, we made minor changes to the abstract and figure 4A. We also included a new supplemental dataset S1 as requested by reviewer #2.

Our responses to the reviewers are in red. The line numbers we have specified correspond to the revised version.

Reviewer #1 (Comments for the Author):

I am satisfied with the revisions.

Reviewer #2 (Comments for the Author):

While the authors make compelling descriptions of their findings in the response-to-reviewer (R2R) document, the abstract remains problematically muddled and overclaiming, which can be fixed with just changes to the text. For the two claims I mention below, care should be taken throughout to avoid overclaiming in other places of the text-but I've highlighted the abstract in detail to explain the problem. I also have a few remaining minor concerns.

- 1) The abstract states " More than five percent of genomes were found to have frameshift mutations in the agr operon. Though most mutations occur only once in the entire species, we observed a small number of recurring mutations evolving convergently across different clonal lineages. Phylogenetic patterns suggested that strains with agr frameshifts were evolutionary dead ends." Together, these statements make the reader think that the authors will show evidence that: (A) that agr mutants cannot transmit to another person; and (B) Any incidence of finding 2 genomes (after genome-wide dereplication) from the same agr type emerges from convergence, rather than identity by descent or recombination.
 - a. In the R2R document, the authors claim they are not intending to claim A. However, the use of the phrase 'dead end' in the abstract does have strong connotations for many readers. A quantitative or relative description of how widespread these mutations are should be used instead.
 - b. In the R2R document, the authors claim they are not intending to claim B. They rightfully note exceptions in which identity-by-descent (IBD) is likely and mention that the point is a matter of degree. However, the abstract explicitly leaves out the notion of IBD and a naïve reader would think the authors think this third possibility is to be ignored.

Both issues could easily be fixed with a simple rewriting. For example, the authors could write "More than five percent of genomes were found to have frameshift mutations in the agr operon. XX of these frameshifts were due to unique mutations found only once, suggesting that *S. aureus* with agr frameshifts are less fit in the long

term. Despite this, we observe cases where the same mutation emerges in distinct clonal lineages, reinforcing the adaptive nature of these mutations in the short term"

We rewrote line 37 to state that we did not see evidence of **long-term** transmission and therefore the frameshifts are **short-lived**. We removed the phrase "dead end". We also included the percentage of uniquely occurring frameshift mutations (52%) in the abstract and in the main text (line 255).

"While 52% of these frameshifts occur only once in the entire species, we observed cases where the recurring mutations evolve convergently across different clonal lineages with no evidence of long-term phylogenetic transmission, suggesting that strains with *agr* frameshifts were evolutionarily short lived"

We also specified that there is no evidence of **long-term** phylogenetic transmission in line 373.

- 2) Figure 4A is now more surprising that 'occurrence' has been better defined (though this should be more explicitly defined in the figure itself -- e.g. '# of CC w same mutation'). The word occurrence, if I am understanding correctly, refers to the number of CCs in which a mutation is found. The convergence here is remarkable and suggestions recombination might be an explanation. If so, it would be useful to color these by *agr* gene (*agrA* may recombine more across CC complexes). Alternatively, if 'occurrence' means 'unique *agr* operon' (a phrase used earlier in the text), this is problematic as an operon can become unique after the acquisition of an *agr* frameshift mutation due to other mutations later occurring on that background.

As stated in line 322 to 329, the number of occurrences of frameshift mutations refers to the number of frameshifts we observed in the dereplicated set of CC22, CC30, CC5 and CC8 genomes. This is also stated in the figure legend (line 1034-1035). Therefore, these are not necessarily "unique" *agr* operons having frameshifts. This is the same dataset that is also used for Fig 4B. We have changed the title of Fig 4A and 4B, as well as the y-axis label of 4A.

- 3) It is not completely clear to me why the authors have discounted the possibility of recombination of a nonfunctional *agr* allele as a cause for its occurrence in multiple CCs.

As stated in line 209 – 212, and 218 – 220, we did not observe instances of shared *agr* gene alleles between different strains apart from the noted exceptions. If non-functional alleles were recombining, we would have observed multiple shared alleles.

We also observed identical mutations across different *agr* groups (Fig 3A,3B), however we do not observe multiple *agr* groups within a CC (except CC45) as would be expected if these mutations were being transmitted by recombination (Fig 1C).

- 4) A supplemental table of every genome analyzed, its CC, *agrA* type, *agr* group, and any frameshifts noted would be a useful tool for the field.

We included supplemental dataset S1 (DatasetS1.xlsx) showing 40,890 genomes analysed, with their NCBI and biosample accessions, ST, CC, *agrA* type, and *agr* group. We also included the number of mutations, their position(s), the type of mutation (insertion/deletion/snp), the effect (frameshift/loss of start/introduction of stop) and the corresponding nucleotide and amino acid change that is caused by the mutation. We also mentioned this in the “Data availability” section (line 604 – 606).

Misc. changes:

Fixed typo in Fig 1A (changed “P1 and P2” to “P2 and P3”)

January 3, 2022

Dr. Timothy D Read
Emory University School of Medicine
Atlanta

Re: Spectrum01334-21R2 (Species-wide phylogenomics of the *Staphylococcus aureus agr* operon reveals convergent evolution of frameshift mutations)

Dear Dr. Timothy D Read,

Thank you for submitting the revised version of your manuscript to Microbiology Spectrum. We have received the comments from both reviewers, and both of them agree that this is an exciting and novel study and should be accepted in its current form. Therefore, it is a pleasure to accept the current version of your manuscript entitled "Species-wide phylogenomics of the *Staphylococcus aureus agr* operon reveals convergent evolution of frameshift mutations" for publication in Microbiology Spectrum. With this email, I am forwarding it to the ASM Journals Department for publication. You will be notified when your proofs are ready to be viewed.

Sincerely,

Gaurav Sharma
Editor, Microbiology Spectrum
<https://sites.google.com/view/sharmaglab/>

Journals Department
Supplemental Material: Accept
Supplemental Dataset: Accept